# Ensemble Machine Learning of Gradient Boosting (XGBoost, LightGBM, CatBoost) and Attention-Based CNN-LSTM for Harmful Algal Blooms Forecasting

**DOI:** 10.3390/toxins15100608

**Published:** 2023-10-10

**Authors:** Jung Min Ahn, Jungwook Kim, Kyunghyun Kim

**Affiliations:** Water Quality Assessment Research Division, Water Environment Research Department, National Institute of Environmental Research, Incheon 22689, Republic of Korea; kjw1128@korea.kr (J.K.);

**Keywords:** harmful algal blooms, Gradient Boosting, attention-based CNN-LSTM, Bayesian optimization, ensemble techniques

## Abstract

Harmful algal blooms (HABs) are a serious threat to ecosystems and human health. The accurate prediction of HABs is crucial for their proactive preparation and management. While mechanism-based numerical modeling, such as the Environmental Fluid Dynamics Code (EFDC), has been widely used in the past, the recent development of machine learning technology with data-based processing capabilities has opened up new possibilities for HABs prediction. In this study, we developed and evaluated two types of machine learning-based models for HABs prediction: Gradient Boosting models (XGBoost, LightGBM, CatBoost) and attention-based CNN-LSTM models. We used Bayesian optimization techniques for hyperparameter tuning, and applied bagging and stacking ensemble techniques to obtain the final prediction results. The final prediction result was derived by applying the optimal hyperparameter and bagging and stacking ensemble techniques, and the applicability of prediction to HABs was evaluated. When predicting HABs with an ensemble technique, it is judged that the overall prediction performance can be improved by complementing the advantages of each model and averaging errors such as overfitting of individual models. Our study highlights the potential of machine learning-based models for HABs prediction and emphasizes the need to incorporate the latest technology into this important field.

## 1. Introduction

Various artificial environmental changes caused by continuous human activities, such as the Four Major Rivers Restoration Project and global climate change, are changing the aquatic environment and increasing the frequency of harmful algal blooms (HABs). Recently, in the Republic of Korea, the problem of water source management has been raised due to the occurrence of HABs in the water source section every summer, and many damages, such as the death of aquatic organisms, are occurring. As a result, the need to preemptively predict and respond to HABs is emerging. Economic losses from HABs over the past 30 years have been estimated at USD121 million. The occurrence, duration, and frequency of HABs are increasing, posing a serious threat to aquatic ecosystems.

The National Institute of Environmental Research (NIER) integrated the water quality forecasting system and the algae warning system in 2020 as a system for managing HABs and provides HABs forecast information to HABs management institutions and the general public so that they can be managed preemptively through HABs forecasting. It is very important to improve the accuracy of HABs prediction by upgrading the HABs prediction technology.

Various studies are being conducted to predict HABs as a method for quickly preparing a policy management plan before or when HABs are expected to occur. Previous studies have focused on improving HABs monitoring technology and raising awareness, and mechanism-based numerical modeling such as the Environmental Fluid Dynamics Code (EFDC) has been considered as an alternative to understanding and mitigating the effects of HABs. Recently, machine learning technology with large data processing capability has been attracting attention, and it is used in various fields such as voice recognition, image analysis, and biological mechanisms. Among various time-series machine learning algorithms, Gradient Boosting and deep learning technologies are being advanced and applied to various topics. Artificial intelligence (AI) methods make significant contributions to the control of a system, determining the decisions to be made about the system, future strategies, and increasing efficiency [1].

Gradient Boosting is generally known to have a higher prediction performance than random forest. Since an ensemble model is constructed using multiple decision trees, it shows high prediction performance. Since the decision tree learns the model by predicting the residual error of the previous decision tree, it has an effect of preventing overfitting. Representatively, there are eXtreme Gradient Boosting (XGBoost) [2], Light Gradient Boosting Machine (LightGBM) [3], and Categorical Boosting (CatBoost) [4].

XGBoost, LightGBM, and CatBoost are all machine learning libraries based on the Gradient Boosting algorithm. XGBoost was developed in 2014 and gained popularity as it performed well on large datasets and won many data science competitions. Since then, it has been developed by adding various functions such as GPU learning and distributed learning through version updates. LightGBM, developed by Microsoft in 2017, has faster speed and lower memory usage than XGBoost and is designed to ensure high speed in large data while ensuring high accuracy even in small data samples. CatBoost was developed by Yahoo in 2017, has strengths in handling categorical variables, and is an optimized algorithm that automatically applies regularization to prevent overfitting and enables fast learning on both CPU and GPU.

Research on deep learning technology began with the RNN (Recurrent Neural Network) model [5], which was structured to calculate the current output value by considering the previous input value. The LSTM (Long Short-Term Memory) model [6] and the GRU (Gated Recurrent Unit) model [7] have also been published. The GRU model, which has a simpler structure than LSTM and is an improved model using a gate to update the state of a memory cell, was introduced to solve the problem that the length of the input sequence and the output sequence are different. The Seq2Seq (Sequence-to-Sequence) model [8] uses two RNN models, an encoder and a decoder, respectively, and was introduced to solve this problem. To overcome the limitations of the RNN model, which uses all information in the input sequence equally, the attention mechanism was developed by Bahdanau et al. [9], a method of extracting information by focusing only on the necessary part of the input sequence and calculating the output value. The transformer model, which further develops the attention mechanism into a multi-head attention form, was introduced by Vaswani et al. [10]. The Temporal Convolutional Network (TCN) model, which combines a 1D-CNN (Convolutional Neural Network) with models such as RNN, LSTM, and GRU, was proposed by Oord et al. [11]. It is a model applied to time-series data prediction using a multi-head attention-based transformer model. Lim et al. [12] proposed a Temporal Fusion Transformer (TFT) model.

Recent research studies on predicting HABs using Gradient Boosting and deep learning techniques have become increasingly prevalent, particularly in the context of time-series data analysis. HABs data, along with various weather and water-quality variables that impact HABs, exhibit a time-series distribution. Kim et al. [13] improved the performance of machine learning models for the early warning of HABs using an adaptive synthetic sampling method. In a study utilizing the Gradient Boosting technique, García et al. [14] employed gradient-boosted regression trees to predict cynotoxin levels. There is also ongoing research employing deep learning techniques, such as the LSTM method, which is particularly effective for time-series analysis and has been widely used for predicting algae (Hill et al. [15]; Liang et al. [16]; Zheng et al. [17]). Li et al. [18] have enhanced HABs prediction by combining ARIMA and LSTM techniques.

Previous studies utilized a single algorithm for HABs prediction. However, in this study, we aimed to combine the Gradient Boosting and deep learning techniques using ensemble methods. Specifically, we developed models using the Gradient Boosting series (XGBoost, LightGBM, CatBoost) and the deep learning series (attention-based CNN-LSTM model) for HABs prediction. Combining diverse models and refining predictions through ensemble techniques can reduce the uncertainty associated with prediction outcomes. Therefore, we sought to combine Gradient Boosting techniques with deep learning. We integrated Gradient Boosting techniques using stacking ensemble methods and combined stacking ensemble techniques with deep learning using bagging methods. The final prediction results were generated from these developed models using bagging and stacking ensemble techniques, and their applicability to HABs was assessed.

## 2. Results and Discussion

### 2.1. Data Selection and Preparation

Water quality, current, and meteorological data from the Nakdong River weir section were used to predict HABs. The water quality data consisted of water temperature (WT) (°C), pH, dissolved oxygen (DO) (mg/L), total nitrogen (T-N) (mg/L), total phosphorus (T-P) (mg/L), and HABs cell counts (cells/mL). The water quality and tide data from the weir section for training were collected from 2014 to 2022 and were directly observed by the National Institute of Environmental Research of the Ministry of Environment. They are open to the public at the Water Environment Information System (http://weis.nier.go.kr/ (accessed on 6 March 2023)). For meteorological data, precipitation (PCP) (mm) data from the Daegu General Weather Station, which are disclosed by the Korea Meteorological Administration, were used.

Since the HABs cell count values ranged from 0 to 1,000,000 cells/mL, they were replaced with log values for the purpose of learning. The log value of the HABs cell count was used as the target, and the prediction result was converted back into the original number of cells. The data that had the highest correlation with Logcyano were water temperature, and the variables that had positive correlations were the month, pH, and T-P (Figure 1).

To preprocess the data, the transform function from scikit-learn’s MinMaxScaler library was used for normalization. In the case of the deep learning model, the data were converted to Tensor data using the Variable function. The preprocessed data is structured as described in Appendix B, consisting of sequences, targets, and goals, and learning and prediction are performed accordingly.

### 2.2. Ensemble Model Development and Prediction

Ensemble learning is a technique that combines predictions from multiple models to obtain more reliable and generalizable predictions. The idea is to average the individual real numbers from different models to reduce the risk of overfitting while maintaining strong predictive performance. Ensemble learning methods include bagging and stacking, which utilize boosting algorithms such as XGBoost, LightGBM, and CatBoost to build models in parallel using random subsets of data (alternative sampling) and aggregate predictions from all models. In this study, as shown in Figure 2, an ensemble model using the Gradient Boosting technique for hyperparameter tuning and prediction results was developed, and the code was configured to derive prediction results.

Hyperparameters of tree-based models, which commonly need to be optimized, include the maximum depth of the tree (max_depth), the number of trees in ensemble learning (n_estimators), and the value of the learning rate (in boosting), as shown in Table 1. Many researchers use GridSearchCV, which was developed by Pedregosa et al. [19] and is included in the scikit-learn8.0 version library, to perform hyperparameter tuning. However, there is a disadvantage that hyperparameter tuning using GridSearchCV takes a long time. Models such as XGBoost, LightGBM, and CatBoost with a large number of hyperparameters require a lot of execution time when tuning using GridSearchCV. Nayak et al. [20] analyzed that hyperparameters were effectively tuned by applying the Bayesian optimization technique to the CatBoost model, and Su et al. [21] applied Bayesian optimization to more effectively select hyperparameters in the XGBoost model. Therefore, this study is based on Bayesian optimization, which is a method of quickly and effectively finding the optimal input value that generates the maximum or minimum function return value for a function whose objective function expression is not properly known, through as few trials as possible. HyperOpt was used for optimization. HyperOpt performs optimization according to the following procedure:(1)Randomly sample hyperparameters and observe performance results.(2)Based on the observed values, the surrogate model estimates the optimal function and confidence interval (=result error deviation = means the uncertainty of the estimation function).(3)Based on the estimated optimal function, the acquisition function calculates the next hyperparameter to be observed and passes it to the surrogate model.(4)Optimization is performed in the order of updating the surrogate model again based on the observed values by performing the hyperparameters passed from the acquisition function.

By repeating steps 2 to 4, it is possible to improve the uncertainty of the alternative model and gradually estimate an accurate optimal function. HyperOpt is an optimization technique in which Bayesian probability improves the posterior probability based on new data. When new data are input, the optimal function is predicted and the posterior model is improved to create an optimal function model. The maximum number of evaluations (max_evals) was set to 50 times, and the final tuned-hyperparameter results are shown in Table 2.

To predict harmful algal blooms (HABs), we generated five bootstrap samples for each XGBoost, LightGBM, and CatBoost model by changing the seed to 0, 1, 2, 3, 4, and 5 using the tuned hyperparameters for each model. The resulting predictions are shown in Figure 3a–c, and the post-processed results using the bagging ensemble method and the stacking ensemble method are shown in Figure 3d,e, respectively. The data used for tuning the hyperparameters were observed from 1 January 2014 to 31 December 2021, and the ensemble prediction period was observed from 1 January 2022 to 31 December 2022.

Prediction accuracy was evaluated using R^2^, MAE, and RMSE (Equations (1)–(3)). For XGBoost, the R^2^ was 0.92, the MAE was 0.2, and the RMSE was 0.4. For LightGBM, the R^2^ was 0.93, the MAE was 0.1, and the RMSE was 0.4. For CatBoost, the R^2^ was 0.89, the MAE was 0.2, and the RMSE was 0.5. All three models produced highly accurate prediction results for HABs. When the bagging ensemble technique was applied to the results of the three models, the R^2^ was 0.92, the MAE was 0.2, and the RMSE was 0.4. When the stacking ensemble technique was applied, the R^2^ was 0.93, the MAE was 0.1, and the RMSE was 0.4. The error size was relatively high for CatBoost around June, and the bagging ensemble technique resulted in a smaller overall error deviation than the stacking ensemble technique.

Although relatively poor prediction results were obtained for CatBoost compared to the other models, the accuracy of the prediction results can vary depending on various factors such as time, location, input data composition, and model. To reduce the uncertainty of the prediction result, we suggest using multiple models with different seeds and performing post-processing with the bagging and stacking ensemble techniques.
(1)$$R^{2}=1-\frac{\sum_{i=1}^{n}{\left(y_{i}-{\hat{y}}_{i}\right)}^{2}}{\sum_{i=1}^{n}{\left(y_{i}-{\bar{y}}_{i}\right)}^{2}}$$
(2)$$MAE=\frac{\sum_{i=1}^{n}\left|y_{i}-{\hat{y}}_{i}\right|}{n}$$
(3)$$RMSE=\sqrt{\frac{\sum_{i=1}^{n}{\left(y_{i}-{\hat{y}}_{i}\right)}^{2}}{n}}$$
where, $y_{i}$ is the observation value, ${\hat{y}}_{i}$ is the prediction value, ${\bar{y}}_{i}$ is the mean of observation value, and n is the number of samples.

In addition to the Gradient Boosting-based prediction model, the results predicted by the deep learning method were also used as an ensemble technique. Although CNN techniques are mainly used for processing images or video data in deep learning, one-dimensional CNN can also be used for time-series analysis because the convolution kernel can be used as a way to automatically extract data features that are not visible in the time direction. Therefore, a sequence was defined as an image, and a network for learning and prediction using the LSTM technique was adopted by extracting the features of the sequence. A network combining Bahdanau attention-based one-dimensional CNN and LSTM was structured as shown in Figure 4 and described as follows:(1)Water quality and algae data are weekly data, so daily data such as temperature are converted to weekly data to ensure that all the data have the same resolution. Here, data preprocessing techniques such as normalization were used to enhance the predictive power of the model. The same preprocessing techniques were applied to XGBoost, LightGBM, and CatBoost.(2)As shown in Figure 4, when the preprocessed data are first input into a 1D CNN of (2, 1) kernel size, the dimension of the sequence is reduced according to the kernel size, and the sequence of the reduced dimension is learned as an input value to LSTM for prediction. The constructed sequence input data at the present (t) are passed through the CNN layer and then input into the Bahdanau attention-based LSTM layer. The number of cyanobacteria cells one week from the present (t + 1) is set as the target for learning.(3)The hyperparameters of the attention-based CNN-LSTM deep learning model, such as hidden size, num layers, dropout rate, and learning rate, were tuned using the data from 2014 to 2021, and the tuned result was applied to learn the bird prediction model using the data from 2022.(4)The process of step (3) was repeated until the verification result was good, and the hyperparameters and weights when the verification result was good were saved. Here, the algae prediction model with stored weights was used to predict future blue-green algae. The model was trained and verified using the same method as described above.

The hyperparameters of the attention-based CNN-LSTM deep learning model were tuned using the same method as the Gradient Boosting technique, and the results are shown in Table 3. ReLU was used for activation, and Adam was adopted as the optimization technique. The attention-based CNN-LSTM model showed an R^2^ of 0.92, MAE of 0.2, and RMSE of 0.4. When the results of the Gradient Boosting (XGBoost, LightGBM, CatBoost) series models and the attention-based CNN-LSTM result were presented as the final predicted value—post-processed with the result of Figure 3d and the bagging ensemble technique—R^2^ was 0.93, MAE was 0.1, and RMSE was 0.4 (Figure 5). It can be seen that the uncertainty of individual models for HABs prediction is reduced when the final prediction results are presented with Gradient Boosting (XGBoost, LightGBM, CatBoost) series prediction results and deep learning attention-based CNN-LSTM prediction results as a bagging ensemble technique, leading to an improved prediction accuracy (Table 4).

Gradient Boosting models (XGBoost, LightGBM, CatBoost) are models that demonstrate strong prediction performance for various datasets, and have high accuracy and fast learning speed. Attention-based CNN-LSTM models are applicable to both image and time-series data, and are able to learn both temporal and spatial patterns. Therefore, by using an attention-based CNN-LSTM model and a Gradient Boosting model (XGBoost, LightGBM, CatBoost) as an ensemble technique for learning and predicting HABs, it is expected that the strengths of each model can complement each other, resulting in an improved prediction performance. Based on the prediction results from various perspectives, it is judged that HABs prediction can be performed by utilizing the strengths of each model, such as recognizing overfitting problems that may occur in specific situations. The algorithm developed in this study can be downloaded from the link in the Appendix A.

## 3. Conclusions

In this study, an algorithm that can predict HABs was developed using an ensemble technique of Gradient Boosting (XGBoost, LightGBM, CatBoost)-based prediction models and deep learning attention-based CNN-LSTM models. The major findings of this study are listed below:(1)Water temperature was found to have the greatest correlation with HABs, and positive correlations were shown in month, pH, and T-P, according to the correlation analysis between the learning data. Since the deviation of the data values for HABs was large, log values were substituted, and data preprocessing was applied with MinMaxScaler normalization.(2)XGBoost, LightGBM, CatBoost models, and attention-based CNN-LSTM models were developed, and optimal hyperparameter results were presented by tuning hyperparameters with Bayesian optimization techniques using observation data from 2014 to 2021.(3)By applying the hyperparameters derived from the Bayesian optimization technique to predict HABs in 2022, the error of the bagged ensemble prediction result of the Gradient Boosting (XGBoost, LightGBM, CatBoost) model was 0.92 for R^2^, 0.2 for MAE, and 0.4 for RMSE, and the error of the stacking ensemble prediction result was 0.93 for R^2^, 0.1 for MAE, and 0.3 for RMSE. Even when predicting with individual methods, the worst results were 0.89 for R^2^, 0.2 for MAE, and 0.5 for RMSE. Therefore, it is judged that the overall prediction performance can be improved by offsetting errors such as those.(4)Not much data have been accumulated for HABs observation even though it has been performed on a weekly basis since 2014. Therefore, it was initially expected that the accuracy of prediction would be low if data-based forecasting techniques were used. However, this study shows that a fairly high prediction accuracy can be achieved by applying the ensemble technique. If future data are accumulated and advanced algorithms are developed, the basis for predicting HABs in advance and utilizing them for policy purposes will be laid.

## 4. Materials and Methods

### 4.1. Study Area

The Nakdong River is the longest river in Korea and one of the four representative rivers. In 2009, the ‘Four Major Rivers Restoration Project’, a large-scale river maintenance project, was launched with the purpose of preventing flood damage, securing water resources, and improving water quality. As a result of this project, eight multi-function weirs were constructed along the Nakdong River: Sangju Weir, Nakdan Weir, Gumi Weir, Chilgok Weir, Gangjeong Goryeong Weir, Dalseong Weir, Hapcheon-Changnyeong Weir, and Changnyeong-Haman Weir. Among them, the algae warning system is operated at the national level for Changnyeong Haman Weir branch due to the presence of a water purification plant used for drinking water (see Figure 6).

The Nakdong River has a gentle slope and many bends, resulting in a slow flow rate and an increase in water temperature every summer. This increase in temperature causes a significant amount of damage from blue-green algae in the downstream area. Since the water system of the Nakdong River directly collects river water and uses it as a water supply source, there is great concern about drinking water quality due to the occurrence of algae blooms.

### 4.2. Gradient Boosting (XGBoost, LightGBM, CatBoost) Method

Gradient Boosting (XGBoost, LightGBM, CatBoost) is a highly reliable technique that is being studied by many researchers in various fields. Dong et al. [22] used the XGBoost model to predict short-term daily rainfall using numerical meteorological forecast data, and Farzinpour et al. [23] applied three hybrid models consisting of boosting-based ensemble learning methods (XGBoost, CatBoost, and LightGBM) for shear strength prediction of squat RC walls, showing high accuracy.

The existing random forest is a prediction method based on decision trees, and each decision tree uses randomly selected variables to partition data and create a predictive model. In contrast, XGBoost, LightGBM, and CatBoost are algorithms based on the Gradient Boosting method. The Gradient Boosting algorithm is a method of improving a predictive model by sequentially adding a series of decision trees and reflecting the residual error of the previous model in the next model. Each decision tree is built by predicting the difference (residual error) between the predicted value and the actual value of the previous model, and it continues to improve the predictive model by reflecting information that the previous model did not predict.

The advantages and disadvantages of XGBoost, LightGBM, and CatBoost are as follows: XGBoost shows high prediction performance for high-dimensional data and sparse data, and works effectively with fast operation speed, scalability, and large data. In addition, it provides various loss functions and flexible cross-validation functions. However, it may have low performance with high memory usage and imbalanced data. LightGBM exhibits fast speed and efficient memory usage, and has excellent processing power for large amounts of data. In addition, it shows high prediction performance for various types of data (categorical variables, numerical variables, and sparse data). However, it can perform poorly on high-dimensional data and lacks the ability to handle it automatically. CatBoost shows great performance in handling categorical variables and provides its own cross-validation function to avoid overfitting problems. In addition, it shows excellent performance in handling outliers or missing values in the data. However, model training is slow, and performance can be poor on high-dimensional data.

### 4.3. Deep Learning (Attention-Based CNN-LSTM) Method

CNN and LSTM are known to be excellent models for processing image and sequence data, respectively. Combining the two models provides the following benefits. It can process image and sequence data simultaneously. CNN shows excellent performance for 2D data such as images, but has limited processing power for sequence data. LSTM, on the other hand, shows excellent performance for sequence data but has limited processing power for 2D data such as images. Therefore, combining CNN and LSTM can process both image and sequence data, which is effective when dealing with various types of data.

LSTM can improve accuracy by learning the features of sequences extracted by CNN. Prediction can be performed by inputting the sequence as an image value, extracting the feature of the sequence with CNN, and inputting the extracted sequence to LSTM. LSTM can be used to consider temporal relationships. LSTM is an algorithm that compensates for the disadvantages of RNN and has the ability to remember past information. Since CNN cannot consider temporal relationships, a model combining CNN and LSTM can perform predictions considering temporal information. Therefore, a model combining CNN and LSTM shows excellent performance in processing image and sequence data, can process various types of data, and can improve prediction accuracy.

Research combining CNN and LSTM is being promoted for the following reasons. Garcia-Moreno et al. [24] used a 1D CNN-LSTM to classify left and right-hand motor imagery EEG, and the algorithm achieved an average accuracy of 87% on the test set. Xu et al. [25] used a 1D CNN-LSTM network to classify MI-EEG data on a five-class epileptic seizure recognition task. Altunay and Albayrak [26] applied CNN-based, LSTM-based, and hybrid CNN-LSTM methods to industrial IoT networks and compared the results. The hybrid CNN-LSTM model could lead to an improved performance of Intrusion Detection Systems. Liang et al. [27] proposed a novel hybrid model with ICEEMDAN and LSTM-CNN-CBAM to forecast gold price. Ahmed et al. [28] proposed an ensemble utilizing the combined predictive performance of three different architectures (1D-CNN-LSTM-GRU model). Zhang et al. [29] predicted the external temperature of an energy pile based on spatial–temporal features using a CNN-LSTM model. Hu and Zhang [30] evaluated rock mechanical parameters using a CNN-LSTM machine learning model.

In this study, attention was combined with the CNN-LSTM model to focus on important parts by learning weights for each time point in the input sequence. The attention technique was developed by Bahdanau et al. [9]. Bahdanau Attention is one of the attention mechanisms first proposed in the field of Neural Machine Translation (NMT). Since it is an encoder–decoder structure that compresses all input sequence information into a fixed-size vector and processes it, the longer the input sequence, the more likely problems will occur. To solve this problem, Bahdanau proposed a mechanism to generate an output using all viewpoint information of the input sequence. The characteristics of Bahdanau Attention are as follows: (1) It generates an output using information from all points in the input sequence, learns the attention weight while generating the output, and identifies the influence of each point in the input sequence on the output. (2) It also concentrates on a certain part of the input sequence to learn what to do, which allows the model to focus on important parts of the input sequence.

### 4.4. Ensemble Prediction Method

Among the various ensemble learning methods, bagging was used to build multiple weak estimators in parallel and combine their results for output [31]. The ensemble approach is widely used in various fields for making predictions because it is simple to implement. Ensemble techniques are methods proposed to demonstrate better prediction performance than a single model by combining multiple prediction models. When prediction is performed using the ensemble technique, overfitting can be prevented by increasing generalization performance and offsetting prediction errors while combining predictions of various models because the same type of base model or different types of base models can be used. A single model may perform poorly in certain situations, but an ensemble model can address this issue by synthesizing the predictions of multiple models, thereby increasing prediction accuracy through the introduction of diversity in decision-making.

Trizoglou et al. [32] presented that an ensemble model of the Extreme Gradient Boosting (XGBoost) framework was successfully developed and critically compared with a Long Short-Term Memory (LSTM) deep learning neural network. The performance of the proposed technique was evaluated as good that XGBoost outperformed LSTM in predictive accuracy while requiring smaller training times and showcasing a lower sensitivity to noise that existed in the SCADA database. Zhang et al. [33] proposed a technique for estimating chlorophyll-a from hyperspectral images, and ensemble techniques such as random forest (RF), Gradient Boosting decision tree (GBDT), Extreme Gradient Boosting (XGBoost), Light Gradient Boosting machine (LightGBM), and Categorical Features and Gradient Boosting (CatBoost) have been applied.

However, the ensemble technique increases the complexity of the model and may increase computational cost when applied to large datasets because various models are performed, but it was adopted in this study because it was judged to have more advantages. Various ensemble techniques have been proposed, such as voting, which is effective for classification; boosting, which is a method of learning by adding weights to incorrectly predicted data; and stacking, which creates a new model by combining several models. Bagging and stacking techniques were selected as ensemble techniques in this study because these methods are known to be effective in regression problems. Additionally, the final prediction result based on the bagging technique was proposed to average the prediction results of the Gradient Boosting series and the attention-based CNN-LSTM model, which is a deep learning series.

## Figures and Tables

**Figure 1 toxins-15-00608-f001:**
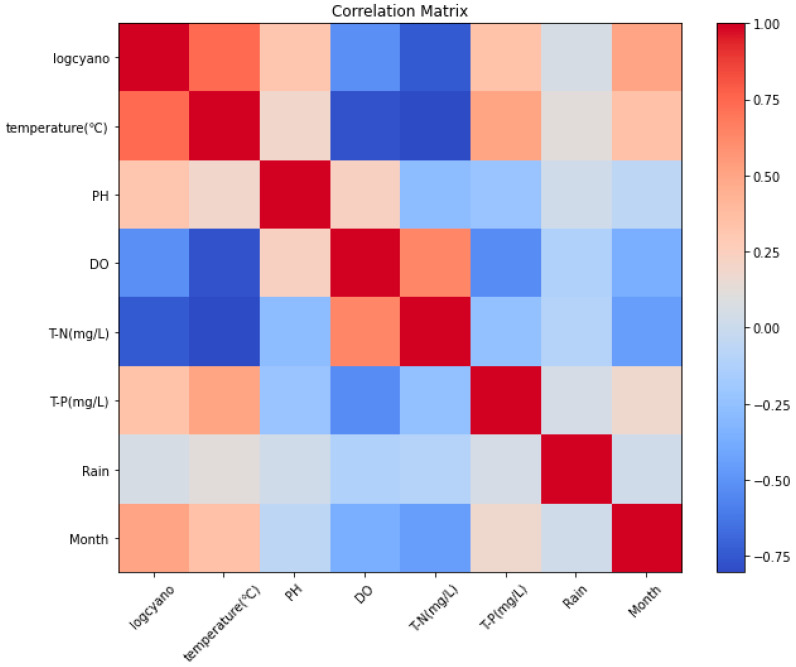
Results of data correlation analysis.

**Figure 2 toxins-15-00608-f002:**
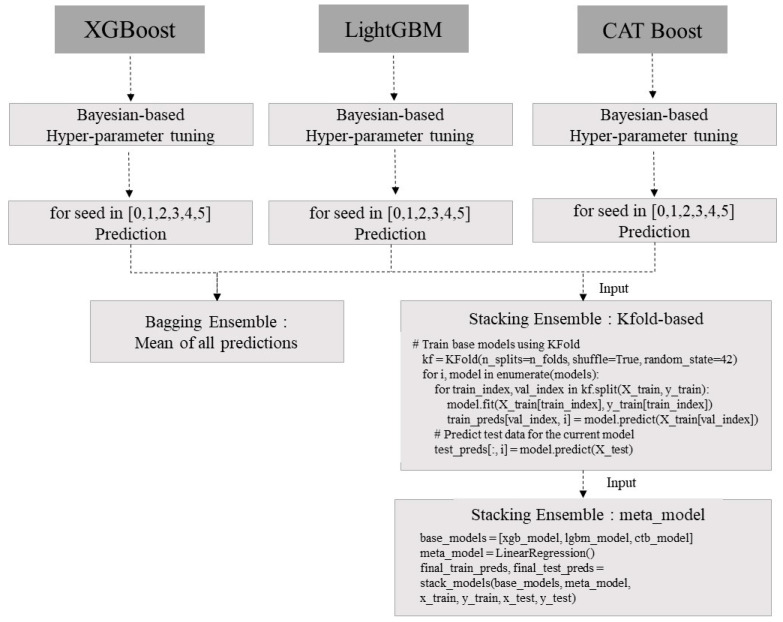
Schematic diagram of ensemble learning using Gradient Boosting technique.

**Figure 3 toxins-15-00608-f003:**
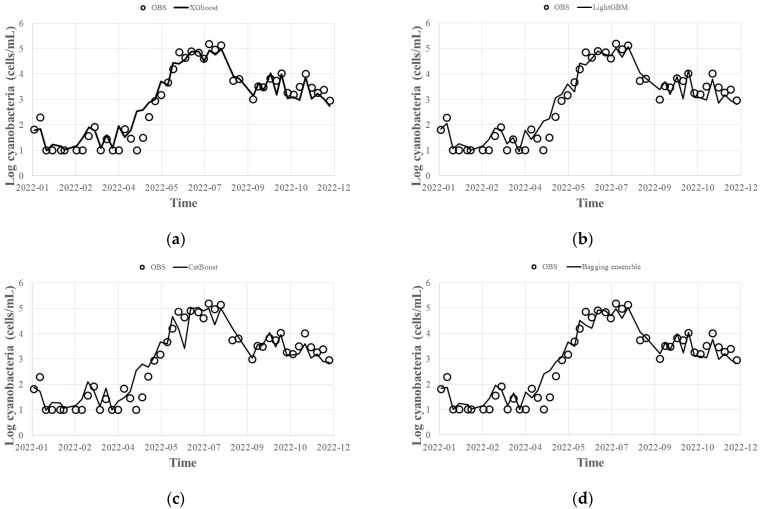
Results of HABs (cells/mL) predicted via Gradient Boosting technique: (**a**) XGBoost; (**b**) LightGBM; (**c**) CatBoost; (**d**) bagging ensemble; (**e**) stacking ensemble.

**Figure 4 toxins-15-00608-f004:**
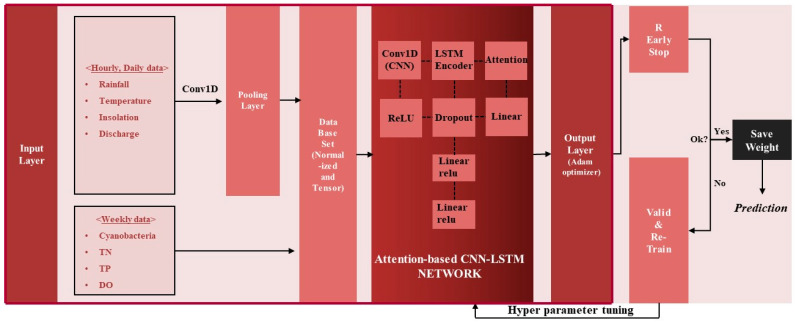
Attention-based CNN-LSTM training and prediction networks.

**Figure 5 toxins-15-00608-f005:**
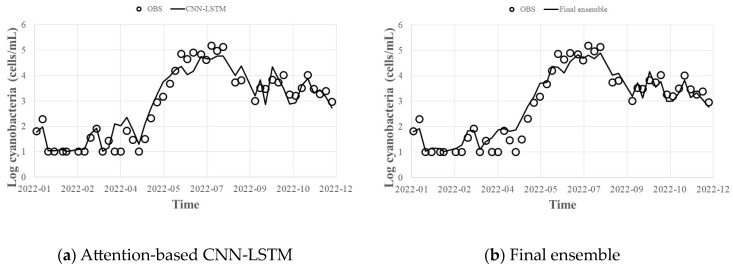
Final HABs (cells/mL) predicted by the attention-based CNN-LSTM technique and the ensemble technique: (**a**) Attention-based CNN-LSTM; (**b**) Final ensemble.

**Figure 6 toxins-15-00608-f006:**
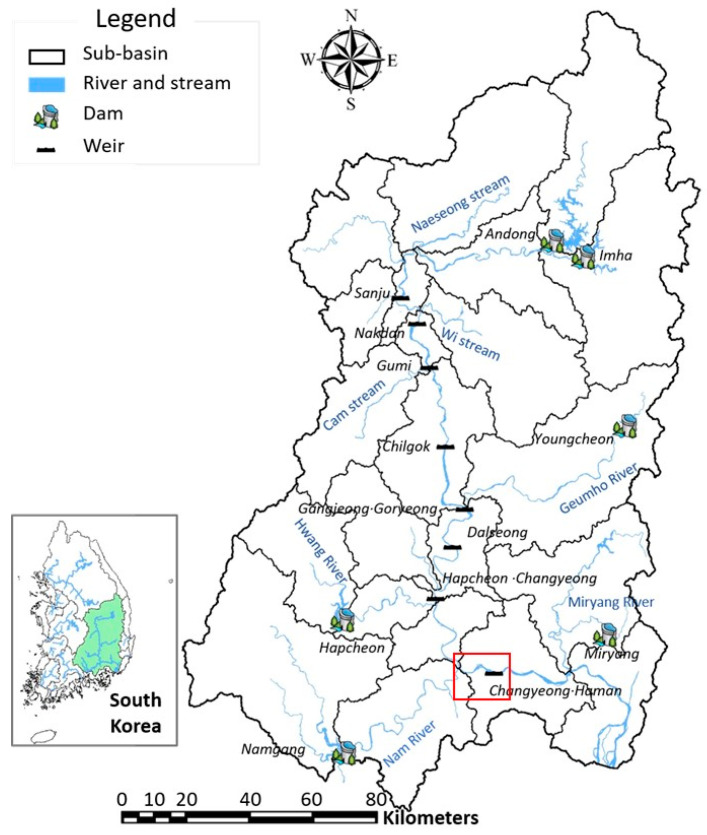
The study area (The red box is the point where monitoring was performed).

**Table 1 toxins-15-00608-t001:** Search range of hyperparameters for developed machine learning models.

Hyperparameters	XGBoost	LightGBM	CatBoost
reg_alpha	1 × 10^−5^, 1 × 10^−4^, 1 × 10^−3^, 1 × 10^−2^, 0.1, 1, 5, 10, 100		-
reg_lambda			-
colsample_bytree	0.5~1.0 (0.1)		-
min_child_weight	250~350 (10)		-
eta	0.1~0.3 (0.1)		-
l2_leaf_reg	-		3~8 (1)
border_count	-		32~255 (10)
colsample_bylevel	-		0.3~0.8 (0.1)
bagging_temperature	-		0~10
min_data_in_leaf	-		1, 5, 10, 20, 30
max_depth	10~25 (1)		3~9 (1)
subsample	0.7~0.9 (0.1)		0.5~1.0
learning_rate	0.01~0.05 (0.005)		
n_estimators	1000~10,000 (10)		
eval_metric	Root Mean Square Error		

**Table 2 toxins-15-00608-t002:** Results of hyperparameters for developed machine learning models.

Hyperparameters	XGBoost	LightGBM	CatBoost
reg_alpha	0	4	-
reg_lambda	7	8	-
colsample_bytree	0.9	1.0	-
min_child_weight	5	1	-
eta	0.1	0.1	-
l2_leaf_reg	-	-	5.0
border_count	-	-	80
colsample_bylevel	-	-	0.7
bagging_temperature	-	-	5.8858097
min_data_in_leaf	-	-	2
max_depth	4	12	3
subsample	0.7	0.8	0.720713
learning_rate	0.05	0.045	0.025
n_estimators	428	574	7600

**Table 3 toxins-15-00608-t003:** Results of hyperparameters for developed CNN-LSTM models.

Hyperparameters	Search Range	Best Parameters
hidden_size	1~128 (1)	4
num_layers	1~3 (1)	2
dropout_rate	0.0, 0.1, 0.2, 0.3, 0.4, 0.5	0.1
learning_rate	0.00001~0.1	0.00568939

**Table 4 toxins-15-00608-t004:** Results of different models.

Model	R^2^	MAE	RMSE
XGBoost	0.92	0.2	0.4
LightGBM	0.93	0.1	0.4
CatBoost	0.89	0.2	0.5
Bagging ensemble	0.92	0.2	0.4
Stacking ensemble	0.93	0.1	0.4
Attention-based CNN-LSTM	0.92	0.2	0.4
Final Ensemble	0.93	0.1	0.4

## Data Availability

Data will be made available on request.

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
