# Peer review of "Ensemble Machine Learning of Gradient Boosting (XGBoost, LightGBM, CatBoost) and Attention-Based CNN-LSTM for Harmful Algal Blooms Forecasting"

_toxins, 2023, doi:10.3390/toxins15100608_

Round 1

Reviewer 1 Report

This manuscript looks at the use of AI and machine learning to improve prediction of HABs. This is a topic that is of interest to many scientists, industries and government officials. There are areas that need to be improved. 

The introduction spends a lot of ink on the models. The authors could put the need in context with some more information on HABs and why their prediction is critical. Additionally, more information like a short description of the different models would be useful for people who don't work within the field and aren't as familiar. 

The methods need improvement. There is information on the models, generally that would fit in the introduction. The methods should focus on how the models apply to your study. Additionally, the methods do not contain enough detail on what you did to allow someone to replicate your experiment. It may seem obvious to the authors, but someone who isn't deeply involved in modeling of this type wouldn't be able to replicate the methods in their location. You also don't discuss anything about the data until the results. This should likely be included in the methods. 

In the results, the authors discuss model development in general terms in great detail. However, it isn't clear how this applies to their prediction of HABs. The graphs and tables are good, but I'm not sure how figure 4 & 6 differ and how table 4 differs from the values reported in the text and why this is important. The authors need to specify how things apply to their results, not just generally. If the different models are relatively indistinguishable, why would you choose one over another? Is this a good fit or not? It looks as though you could just set temperature and TP limits and get pretty good prediction, why would you want to utilize all the computing power? In general, make sure you spell out all acronyms the first time they are used in text. Be consistent in how you refer to things. For example, at different places in the manuscript, you refer to total phosphorus as TP and T-P. 

The conclusions do not seem supported by the results, but this may have more to do with the lack of specificity in the rest of the manuscript. The final conclusion seems to suggest that the authors were surprised by how well their models predicted HABs, but I am not sure why. 

Overall the English is good. There are a few sentences that would be improved with some additional editing. For instance 199-202 starting The performance of the proposed technique was evaluated as good that... This needs to be reworded.

The paragraph starting at line 207 contains many run on sentences and could also stand editing for sentence structure. 

Author Response

I have attached the response to reviewer.

Reviewer 2 Report

In this study, two types of machine learning-based models, i.e. gradient boosting models and attention-based CNN-LSTM models are performed for HABs prediction.

As an application of machine learning algorithms in this field, it has certain application value. However, in the introduction, the author did not provide enough reference on the current study status and shortcomings on HAB prediction, but instead spent too much space on algorithm introduction, which deviated from the research topic of this article, and seemed that the purpose is to introduce algorithms instead of HAB prediction. In addition, data is the key to machine learning, but the data preprocessing, standardization, and how to input are not explained unclear. I hope the author can supplement the above two points, namely, related reference and data preprocessing.

Author Response

(The authors gave the same response as above.)

Round 2

Reviewer 1 Report

Thank you for taking the time to address my concerns. The edits clarified things well. I have no other concerns.